# Are Ratings of Perceived Exertion during Endurance Tasks of Predictive Value? Findings in Trunk Muscles Require Special Attention

**DOI:** 10.3390/jfmk9040180

**Published:** 2024-09-27

**Authors:** Christoph Anders, Lena Simone Mader, Max Herzberg, Christin Alex

**Affiliations:** Division of Motor Research, Pathophysiology and Biomechanics, Experimental Trauma Surgery, Department for Hand, Reconstructive, and Trauma Surgery, Jena University Hospital, Friedrich-Schiller-University Jena, 07740 Jena, Germany; lena.mader@uni-jena.de (L.S.M.); max.herzberg@uni-jena.de (M.H.); christin.alex@uni-jena.de (C.A.)

**Keywords:** abdominal muscle, back muscles, submaximal load level, fatiguing task, human

## Abstract

**Background**: Subjective rating scales of perceived exertion are often used to quantify effort levels during various endurance exercises, particularly submaximal tasks. The aim of the current study was to determine whether predictive conclusions can be drawn from perceived exertion levels surveyed at the start of defined submaximal endurance tasks. **Methods**: In this study, healthy participants performed a 10-min endurance task at 50% of their upper body weight, targeting either the back muscles (n = 47, 24 women) or abdominal muscles (n = 32, 17 women). At the end of each minute, participants were asked to rate their perceived exertion (RPE) using the 14-points Borg Scale. Based on their initial and final RPE levels, and for each muscle group separately, participants were divided into subgroups reflecting low (good start/good end) and high (bad start/bad end) strain levels. These values were then compared over the duration of the exercise. Comparisons of RPE levels between subgroups were made using the Mann-Whitney U-test for independent samples, with Bonferroni-Holm correction to account for multiple comparisons. **Results**: Overall, strain levels increased throughout the duration of the exercise. For the abdominal muscles, the difference between the two RPE groups remained constant over time: participants with good start/end ratings consistently showed different strain levels from those with bad start/end ratings, regardless of whether the grouping was based on initial or final exertion levels. In contrast, for the back muscles, the initial grouping showed a crossover in strain values: by the end of the task, participants in the good start group tended to report higher strain than those in the bad start group. No differences were found in initial RPE values when the grouping was based on final exertion levels. **Conclusions**: For endurance tasks involving the abdominal muscles, initial strain levels have strong predictive value, whereas this is not the case for the back muscles. Because back muscles are frequently loaded, continuous monitoring of RPE levels is necessary to prevent unexpected task failure, as initial RPE values are not predictive. In contrast, RPE values of 11 or higher on the 14-points Borg scale predict complete exhaustion or even premature task failure with high certainty for abdominal muscle exercises, while lower RPE levels indicate that exercise intensity can be increased.

## 1. Introduction

Whether during leisure activities or professional tasks, submaximal endurance requirements are ubiquitous. However, accurately quantifying them is often challenging. For instance, muscle fatigue can be measured using changes in amplitude and frequency during endurance exercises [1], although neither parameter is equally suitable. The appropriateness of amplitude or frequency changes for assessing muscle fatigue depends on the specific muscle region being examined [2,3]. Moreover, even when quantifiable changes in these physiological parameters are observed, no standardized values exist to differentiate between mild and severe fatigue. In addition to gender-related differences [4], regional differences [5] must also be taken into account, though their assessment remains inconclusive.

Beyond these factors, quantifying muscle fatigue based on physiological parameters requires specialized instrumentation and often involves complex data analysis, as technical or physiological artifacts [6] must be identified and corrected to draw reliable conclusions. Such analyses are typically performed offline, i.e., retrospectively, highlighting the need for research and development to enable real-time physiological characterization of muscle fatigue.

Given these challenges, attempts were made early on, particularly in workplace-related studies, to obtain valid, time-synchronized subjective assessments of strain using rating scales. Various ratings of perceived exertion (RPE) scales were developed and validated for this purpose. Depending on the scale used, exertion levels are typically rated from 0 (no exertion) to either 10 or 100 (maximum exertion) [7,8,9]. An exception is the Borg Scale, introduced in 1970, which ranges from 6 (very, very light) to 20 (very, very strenuous) [10]. Originally designed for quantifying whole-body demands, the Borg Scale can also be applied to load situations without cardiovascular stress, such as resistance or endurance exercises involving different body regions [11].

Such data are particularly valuable because they can accurately describe the individual’s level of effort or fatigue and indicate impending exhaustion or task cessation [12]. However, a valid question arises regarding the extent to which familiarity with such scales or experience with general or specific physical exertion influences the results [13,14,15]. Additionally, it remains unclear whether perceived exertion values recorded at the beginning of a defined endurance task hold predictive value for task completion or the exertion expected at its conclusion. This question formed the basis of the current study.

The objective of the study was to determine whether RPE values collected during submaximal endurance tasks of a defined duration could predict performance or exertion at the end of the task. Given that most activities occur while sitting, standing, or during movement, the study was conducted in an upright position, with participants carrying 50% of their upper body weight for 10 min. Perceived exertion was measured using the 14-points Borg scale [10], and participants were categorized into low and high-exertion groups at both the beginning and end of the endurance task. 

The study consisted of two substudies: one focusing on the back muscles and the other on the abdominal muscles. Based on the frequent use of back muscles in daily life, we hypothesized that perceived exertion ratings would be more predictable for the back muscles, whereas this predictability was not expected for the abdominal muscles. Furthermore, we anticipated higher exertion ratings for the abdominal muscles [16] due to the greater strength capacity of the back muscles [17].

## 2. Materials and Methods

For the study, two cohorts of healthy individuals were assessed during a time-limited endurance test. In Substudy I, the focus was on the back muscles, while in Substudy II, the abdominal muscles were examined.

### 2.1. Participants

In Substudy I, 47 healthy individuals (24 women; anthropometric data, see Table 1) were examined, while 32 healthy individuals (17 women; anthropometric data, see Table 1) participated in Substudy II. Participants were recruited from the general public through printed and online advertisements.

The presented results are part of two larger studies investigating the reliability characteristics of Surface Electromyography (sEMG) measures during endurance tasks. The sample size for the respective studies was calculated for repeated measures, requiring 34 subjects to achieve sufficient power for two-sided tests (effect size = 0.5, power = 0.8).

The following inclusion and exclusion criteria were applied to all participants. Inclusion criteria: written informed consent, Body mass index (BMI) < 30 kg/m^2^, height < 190 cm (due to device limitations), and age between 25 and 50 years. Exclusion criteria: acute back pain, previous spine or trunk surgeries, clinically significant orthopedic conditions (e.g., scoliosis), and intense physical activity during leisure time (>two training sessions or hours of activity per week).

Both substudies were reviewed and approved by the Ethics Committee of the Friedrich-Schiller-University Jena (approval numbers: 2021-2373-BO, 2021-2373_1-BO), ensuring they met ethical standards for human research in accordance with the current version of the Declaration of Helsinki.

### 2.2. Investigation

All subjects were tested using the computer-assisted test and training device system (CTT) Centaur (BfMC Leipzig, Germany). In the device, the lower body is securely fixed, while the upper body remains free to move (Figure 1). The device applies whole-body tilts, allowing subjects to maintain their upright posture while being tilted. 

To ensure correct adherence to upright posture, the device is equipped with a biofeedback monitor positioned directly in the subject’s line of sight. The monitor displays a movable point, which deviates from the center of a crosshair if any force is applied to the shoulder harness positioned over the subject’s shoulders (see Figure 1). This setup allowed for continuous monitoring and, if necessary, correction of posture throughout the entire task.

The CTT Centaur can be tilted at any angle between 0° and 90°, with rotation angles ranging from +180° to −180°, allowing the application of forces between 0% and 100% of the upper body weight (UBW) in any desired direction. During the exercises, subjects kept their arms crossed in front of their chests. For the endurance task, the load was always applied at a 30° tilt angle, corresponding to 50% UBW, regardless of the substudy. Before the endurance test, participants performed a submaximal warm-up of their trunk muscles with short-term tilts at various angles in the device.

In Substudy I, the load was applied at a 0° rotation angle, corresponding to a forward tilt, while in Substudy II, the load was applied at a 180° rotation angle, corresponding to a backward tilt. The endurance task involved applying an isometric load at 50% UBW for a total duration of 10 min. For this analysis, only data from subjects who successfully completed the endurance task were included.

During the endurance task, perceived exertion was requested using the 14-points Borg scale [10] after each elapsed minute. The Borg scale ranges from 6 to 20 and provides descriptive labels for its numerical values. A score of 6 corresponds to “very, very light”, while a score of 20 represents “very, very strenuous”.

For each RPE query, the elapsed time was announced, allowing participants to anticipate the defined end of the endurance task for motivational purposes. Verbal encouragement was provided to all participants and intensified if deemed necessary to prevent premature task failure. Additionally, RPE values were always collected by an investigator of the opposite sex to minimize the potential for motivational bias related to gender [18].

### 2.3. Definition of Subgroups

To assess the impact of the endurance task on individual RPE levels, participants in each substudy were divided into extreme groups based on their perceived exertion levels. This division was done separately for each substudy and was strictly based on the RPE values [10] reported at the beginning and the end of the endurance task. Groups were labeled as “good start” and “bad start” based on the RPE ratings after the first minute of the task and as “good end” and “bad end” according to the RPE ratings at the end.

Group allocations were made with the aim of maintaining similar group sizes within each subgroup and were based on the reported RPE values (Table 2).

### 2.4. Data Analysis and Statistics

The study also included the application of submaximal load levels and the measurement of voluntary maximum trunk extension and flexion contractions (MVC: the maximum of three extension and flexion tasks performed in the device, i.e., in an upright position). Surface EMG (sEMG) recordings of representative trunk muscles were taken for further analysis. The examined muscles included the rectus abdominis (RA), internal oblique (OI), external oblique (OE), multifidus (MF), and longissimus (LO) muscles. sEMG measurements were conducted according to SENIAM recommendations (www.seniam.org, accessed on 1 September 2021), which cover skin preparation, electrode placement, signal amplification, AD conversion, and analysis.

In the present analysis, the sEMG amplitudes at the beginning of the endurance task are provided as normalized values based on the MVC and labeled as relative load levels.

For the statistical analysis of RPE values, the nonparametric Mann-Whitney U-test was used due to the small sample sizes in the subgroups. To prevent alpha error accumulation from multiple tests, *p*-values were adjusted using the Bonferroni-Holm correction [19]. The global significance level was set at *p* < 0.05. Additionally, effect sizes were calculated. Since this study forms part of two larger studies, no separate sample size calculation was performed.

## 3. Results

### 3.1. Relative Load Levels of Muscles

The relative load levels of the examined trunk muscles did not show any systematic side differences across the entire group. Additionally, systematic differences in normalized amplitude values were only observed bilaterally between RA and OE, the OE and MF, as well as between OI and OE on the right side, with the OE consistently displaying highest values (Figure 2). This indicates that at the start of the isometric endurance task, abdominal and back muscles experienced similar strain levels.

For the subgroups classified as “good start/end” and “bad start/end”, MVC-normalized amplitudes for the RA and OI showed a tendency toward higher relative amplitude levels in the “bad” groups. However, no significant differences or relevant effect sizes were identified. At the beginning of the endurance task, no detectable differences in relative amplitude levels were found between subgroups for any of the other muscles (see Figure 2).

### 3.2. Perceived Exertion

For the abdominal muscle endurance test, significant differences with correspondingly high effect sizes were observed between the “good start/end” and “bad start/end” RPE values, regardless of whether the grouping was based on RPE values at the beginning or the end of the endurance task (Table 2, Figure 3).

In contrast, for the back muscle endurance test, significant differences between the “good start” and “bad start” RPE values were observed only until the 4th minute. After that, the differences were no longer significant. Notably, effect sizes suggest a potential crossover phenomenon, with the “good start” group reporting higher RPE values than the “bad start” group by the end of the endurance test (see Table 3). When comparing “good end” versus “bad end” groups, based on RPE values at the end of the task, there were no systematic differences in RPE start levels between the subgroups (neither in the test statistics nor in the effect sizes). However, from the 4th minute onwards, significant differences emerged, with higher RPE ratings for the “bad end” group (Figure 3).

## 4. Discussion

The present study investigated the extent to which perceived exertion levels at the beginning or the end of a time-limited endurance task for trunk muscles can be generalized and whether values obtained at the beginning have predictive value.

In general, MVC-normalized amplitude levels of the examined trunk muscles, determined at the beginning of the load, did not show systematic differences except for OE, which consistently exhibited the highest values. The relative amplitude values for the abdominal and back muscles were comparable, with the OE showing the highest levels of 28% on the left and 31% on the right. All other values ranged from 20% (OI left) to 24% (LO left), indicating similar relative load levels.

The observed values were within the expected range for healthy individuals, assuming a physiological strength reserve of approximately 100% of the UBW [17,20]. Although maximum strength capacity for trunk flexion in healthy untrained subjects is expected to be about 30% lower than for extension [17], the normalized values at the beginning of the task showed comparable relative load levels between abdominal and back muscles. Given that abdominal and back muscles likely have different fiber compositions [21,22], with abdominal muscles having a higher proportion of type II fibers compared to back muscles [21,23,24,25,26], it is anticipated that abdominal muscles would exhibit a nonlinear amplitude characteristic above the applied load level. This expectation is supported by existing studies [27,28] and aligns with the current results.

The results regarding differences in perceived exertion were surprising, as abdominal and back muscle endurance tasks exhibited completely different characteristics. The localization of the load and the resulting functional load situation clearly play a significant role. For the abdominal muscles, it appears that the timing of the RPE query—whether at the beginning or end of an endurance task—does not impact the detection of distinctly different RPE levels throughout the task. Regardless of the reference time point, markedly different RPE levels were identified for the defined RPE subgroups. The observed differences of 3–4 RPE items, or 20–28% on the 14-points Borg scale, are to be considered substantial. Consequently, assessing a relevant exertion level at the beginning (with a value of 11 corresponding to a “light” stress level) seems to be predictive. Both linearly interpolated RPE curves showed similar inclines (see Figure 3), indicating that the gap between the two RPE levels remained almost constant over the 10-min test period, regardless of the reference time point.

The situation was markedly different for the functional endurance task of the back muscles (see Figure 3). In this case, grouping based on initial RPE values resulted in a crossover of the curves, with the “good start” subgroup tending to show higher RPE values than the “bad start” subgroup by the end of the task. This phenomenon cannot be explained by the available physiological data, as relative amplitudes were virtually identical between the subgroups (see Figure 2). The most plausible explanation for this crossover might be inexperience with such load situations. The literature suggests that inexperienced participants may tend to underestimate the level of demand at the beginning [29]. However, all participants were inexperienced in exercise execution, including those in Substudy II for the abdominal endurance task. We also assessed physical activity during leisure time but found no indications of potential confounders.

Interestingly, the “good start” subgroup consisted almost entirely of women. According to the literature, women may exhibit superior endurance capacity during isometric fatiguing tests compared to men [30], but perceived exertion during the endurance task seemed to deviate from that of men. There are conflicting findings in this area [31], so a definitive statement cannot be made with absolute certainty. While differences in competitiveness between men and women have been reported [32], the collected data do not provide conclusive evidence to explain these findings. To minimize potential motivation-related bias [18], study supervisors of the opposite sex questioned the RPE values and provided corrective and motivational support during the endurance task.

For the subgroup formation based on the different RPE values at the end of the task, no indications could be found at the beginning of the load, either for the RPE or for the relative amplitude values, that would have had predictive value. However, the proportion of female participants in the “good end” group was higher than that of men. Similarly, no difference in sporting leisure activity could be determined for the good and bad subgroups at the end, as was the case for the subgroups at the beginning.

### Limitations

The study was conducted in an experimental setting, so the results should be interpreted with caution when applying them to everyday situations. Participants were naive to both the testing environment and the RPE scale used, which may have influenced the data.

The small sample sizes of the extreme groups, based on both initial and final RPE values, limit the generalizability of the findings. Additionally, excluding dropouts from the analysis may have introduced bias.

Furthermore, the limited scale used in the study exhibits a ceiling effect, which should also be considered when interpreting the results.

## 5. Conclusions

The present study clearly demonstrates that predictive statements about expectable RPE values at the end of a time-defined endurance task can be made for the abdominal muscles based on initial RPE values. In contrast, such predictive conclusions cannot be drawn for back muscles. This has practical implications for testing, training, and occupational applications. For tasks involving back muscles, initial RPE levels should be regularly updated during task execution to prevent unexpected task failure. Conversely, RPE values for abdominal muscle loads are predictive of good task performance and also premature failure or task completion.

### Practical Implications

Based on the presented data, clear threshold values can be defined for abdominal muscle exercises using the Borg scale. A value of ≥ 11 on the 14-points Borg scale indicates a risk of premature termination or exhaustion, while lower values suggest that exercise intensity can be increased. This information is useful for specific training or rehabilitation processes. However, the data do not provide guidance on how load intensity or duration might be adjusted, and further studies are needed to address this. In contrast, such predictive statements cannot be made for back muscle exertion. Therefore, exercises involving back muscles require continuous monitoring of perceived exertion to prevent premature termination or complete exhaustion.

## Figures and Tables

**Figure 1 jfmk-09-00180-f001:**
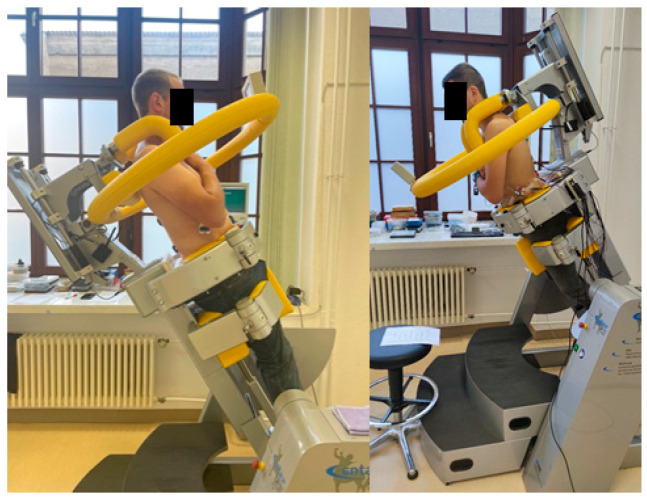
Participants were positioned in the device at a 30° tilt angle (50% of upper body weight). (**Left**): backward tilt (Substudy II), (**right**): forward tilt (Substudy I). Please note that the feedback monitor for posture control is just in front of the participant.

**Figure 2 jfmk-09-00180-f002:**
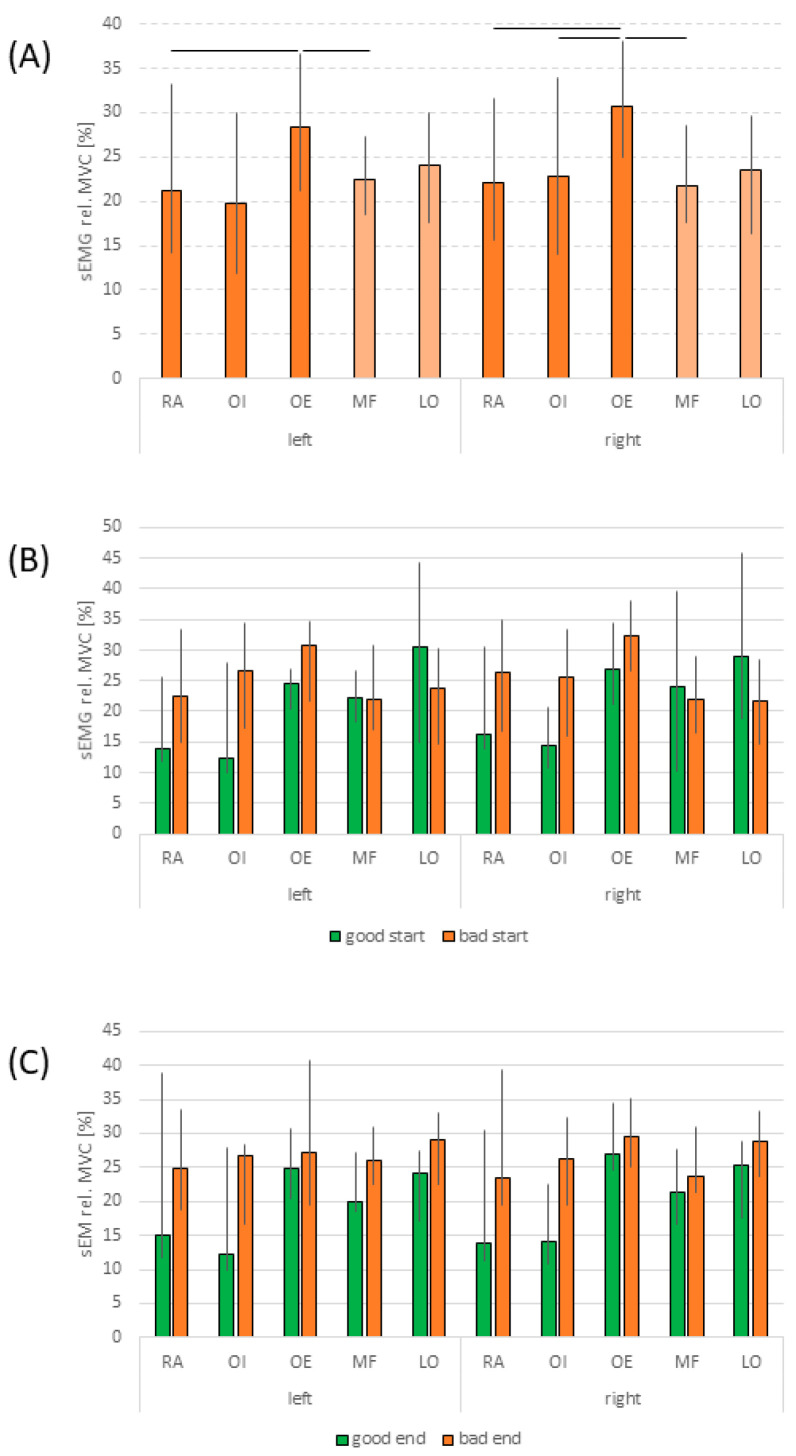
Normalized sEMG amplitudes for the investigated muscles at the beginning of the endurance task: (**A**) MVC-normalized sEMG amplitudes of abdominal and back muscles at a load of 50% upper body weight for the entire group. Horizontal bars indicate significant differences (*p* < 0.05) between values. (**B**) Data for “good start” and “bad start” subgroups. (**C**) Data for “good end” and “bad end” subgroups. All data are displayed as median values with upper and lower quartile ranges. LO: longissimus muscle; MF: multifidus muscle; OE: external oblique muscle; OI: internal oblique muscle; RA: rectus abdominis muscle; sEMG: surface EMG.

**Figure 3 jfmk-09-00180-f003:**
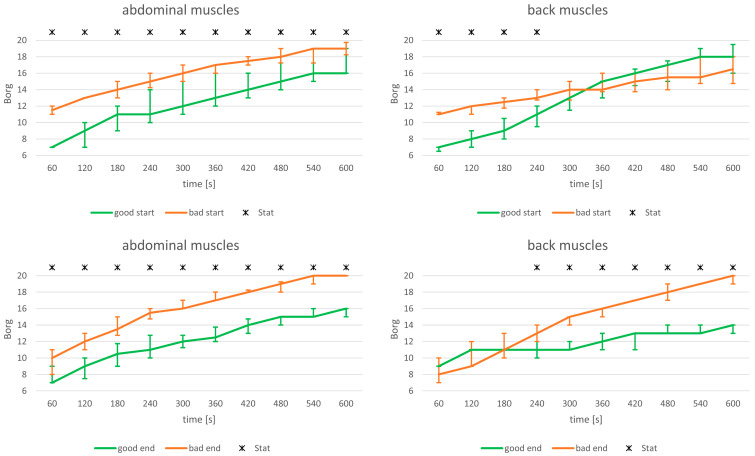
Differences in ratings of perceived exertion between the “good start/end” and “bad start/end” subgroups during a 10-min isometric endurance test of abdominal and back muscles. Load level: 50% of upper body weight. Asterisks indicate significant differences between subgroups (*p* < 0.05), as determined by the Mann-Whitney U-test with Bonferroni-Holm correction (global significance level: *p* < 0.05).

**Table 1 jfmk-09-00180-t001:** Anthropometric characteristics of participants (separately for both Substudies).

	Parameter	Age [years]	Height [cm]	Weight [kg]	BMI [kg/m^2^]
Study I					
Female (n = 24)	MW	34.8	167.6	65.3	23.3
	SD	7.5	7.0	7.1	2.7
	Min	25	158	79.4	19.4
	Max	47	186	51.3	27.9
Male (n = 23)	MW	31.7	181.0	77.9	23.8
	SD	5.8	6.0	8.3	2.1
	Min	49	193	59.8	20.0
	Max	25	168	94.3	27.0
Statistics	*p*-value	0.12	<0.01	<0.01	0.52
	ES	0.46	2.05	1.63	0.19
Study II					
Female (n = 17)	MW	35.3	167.2	65.4	23.3
	SD	10.0	7.0	12.2	3.4
	Min	24	154	41	16.0
	Max	48	180	91.4	29.2
Male (n = 15)	MW	41.3	179.6	81.6	25.3
	SD	6.6	5.2	8.1	2.1
	Min	24	165	70.8	21.1
	Max	52	190	103.0	31.5
Statistics	*p*-value	0.06	<0.01	<0.01	0.06
	ES	0.71	2.02	1.56	0.70
Study I vs. II					
Female	*p*-value	0.84	0.88	0.98	0.97
	ES	0.06	0.05	0.01	0.01
Male	*p*-value	<0.01	0.48	0.19	0.04
	ES	1.55	0.24	0.44	0.71

BMI: body mass index; ES: effect size; MW: mean value; SD: standard deviation; Min: minimum value; Max: maximum value.

**Table 2 jfmk-09-00180-t002:** Subgroup definitions and distribution statistics based on RPE values.

	Mean	SD	Median	u. Q.	l. Q.	Max	Min
Substudy I							
Good start (n = 11)	6.7	0.5	7.0	7.0	6.5	7	6
Bad start (n = 12)	11.3	0.5	11.0	11.0	11.25	11	12
Good end (n = 13)	12.8	2.0	14.0	14.0	13.0	14	7
Bad end (n = 13)	19.5	0.5	20.0	20.0	19.0	19	20
Substudy II							
Good start (n = 9)	6.8	0.4	7.0	7.0	7.0	7	6
Bad start (n = 14)	11.6	0.7	11.5	11.0	12.0	13	11
Good end (n = 10)	15.6	0.5	16.0	17.0	16.0	16	15
Bad end (n = 8)	20.0	0.0	20.0	20.0	20.0	20	20

SD: standard deviation, u. Q.: upper quartile, l. Q.: lower quartile, Min: minimum value, Max: maximum value, RPE: ratings of perceived exertion.

**Table 3 jfmk-09-00180-t003:** Effect Sizes for the Comparison of RPE values: Good vs. Bad for the Reference Times “Start” and “End”.

Time [s]	60	120	180	240	300	360	420	480	540	600
Abdominal muscles										
Start	0.86	0.85	0.84	0.77	0.59	0.56	0.53	0.50	0.49	0.50
End	0.50	0.69	0.70	0.81	0.85	0.85	0.86	0.85	0.86	0.90
Back muscles										
Start	0.90	0.86	0.80	0.60	0.27	0.01	0.12	0.17	0.21	0.21
End	0.20	0.01	0.24	0.63	0.87	0.91	0.92	0.92	0.92	0.93

Assessment of effect size values: <0.3 small, 0.3–0.5 medium, >0.5 large. RPE: ratings of perceived exertion.

## Data Availability

All data are available upon request.

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
