# Peer review of "Are Ratings of Perceived Exertion during Endurance Tasks of Predictive Value? Findings in Trunk Muscles Require Special Attention"

_jfmk, 2024, doi:10.3390/jfmk9040180_

Round 1

Reviewer 1 Report

Comments and Suggestions for Authors

i suggest, if the authors agree, to change the title for: Predictive values of ratings for perceived exertion during endurance tasks? As study with trunk muscles

in the methods section please add the (M±SD) values, the minimum and maximum age of participants;

if possible, a comparison to test if there are any statistical differences between abdominal and back muscles;

in the conclusions section authors state that:  its demonstrated that it is possible to make predictive statements about the expected RPE. which practical apllications can ne done in the future?

Comments on the Quality of English Language

Minor editing of English language required

Author Response

Comments and Suggestions for Authors

i suggest, if the authors agree, to change the title for: Predictive values of ratings for perceived exertion during endurance tasks? As study with trunk muscles

Answer

As the other reviewers were happy with the Title we would leave it as it stands.

in the methods section please add the (M±SD) values, the minimum and maximum age of participants;

Answer

All required parameters are now included

if possible, a comparison to test if there are any statistical differences between abdominal and back muscles;

Answer

In (new) figure 2 already relative amplitudes (with respect to MVC levels) were compared. The other comparisons were conducted between the god/bad start and end groups.

in the conclusions section authors state that:  its demonstrated that it is possible to make predictive statements about the expected RPE. which practical apllications can ne done in the future?

Answer

We have added a respective application

Reviewer 2 Report

Comments and Suggestions for Authors

Thank you for the opportunity to review this work. My main concern is the usefulness and potential application of these findings.  I don't see what the meaning of this study is, what is the novelty  that can drive the field vertically. This does not make me inclined to endorse the publication but I want to give the authors the opportunity to clarify this (fundamental) point. I choose the major revisions, for now, for the general clarity and correctness of the data analyses. However, the following points must be addressed.

What improvements do your results bring? How? The authors should propose practical applications of  the main result of the work (i.e. that it is possible to predict RPE values ​​for the abdominal muscles based on the initial RPE values). Please note that formally the work needs an extensive revision. First of all, an english native speaker revision in requested. Here some additional hints:

- Abstract should end with a concise description about the possible usefulness of the proposed data.

- Please change the word "prognostic" with "predictive" throughout the manuscript.  

- The group names (good/bad /start/end) are generic, inadequate and need to be more clearly related to the scale they refer to. Please edit throughout the text.

-Line 44-45: do you mean in real time?

- Line 55: only for endurance? not also for resistance exertion?

- Line 73-74: Add at least two references.

- Methods: for the exclusion criteria, wouldn't  be better to indicate any permanent or transitory condition that may affect the ability to carry out the proposed exercises?

- Table 1: Each acronym used should have its extended form below the table

- Investigation: please add a descriptive/representative image of the experimental set up and of the proposed exercises.

- Figure 3 is unnecessary.

-Line 197-199: Why?

-Line 203-206: Please revise the statement in a less absolutist form.

-Line 211-212: Is this percentual universal? Please revise the statement.

- Line 217-218: It may not be hypothetical but, based on the proposed references (both including 39 subjects, same cohort?), it is not written in stone either! Please modify the sentence by specifying that the proposed concept is in line with previous studies. Also, please, if possible, it would be better to avoid too many self-citations.

- Line 219-221: Please widely modify.

- Line 222: delete "evidently".

- Line 230: Please add a reference to the text parts which report it.

- Line 241: is this reported in the methods section? 

- Weren't you aware of the issue? Why didn't you include a certain level of experience of the participants among the inclusion criteria to avoid this bias? This is a gross methodological error!

- Line 243-245: this is true for isometric, but not isotonic, task (according to the mentioned reference). Please reformulate.

- Line 247-249: Is this data universally generalizable? Rephrase the sentence please.

-Line 250-252: Where is this mentioned in the methods section? Why of the opposite sex?

Comments on the Quality of English Language

The work is formally inadequate and needs to be  revised by a native speaker.

Author Response

Thank you for the opportunity to review this work. My main concern is the usefulness and potential application of these findings.  I don't see what the meaning of this study is, what is the novelty  that can drive the field vertically. This does not make me inclined to endorse the publication but I want to give the authors the opportunity to clarify this (fundamental) point. I choose the major revisions, for now, for the general clarity and correctness of the data analyses. However, the following points must be addressed.

Answer

We have completely revised the manuscript to improve language.

What improvements do your results bring? How? The authors should propose practical applications of  the main result of the work (i.e. that it is possible to predict RPE values ​​for the abdominal muscles based on the initial RPE values). Please note that formally the work needs an extensive revision. First of all, an english native speaker revision in requested. Here some additional hints:

Answer

During submaximal endurance tasks premature failure is not due to physiological, but psychological, i.e. motivational reasons, i.e. the perceived exertion during such tasks. Therefore it makes sense to query such ratings during endurance tasks, to predict failure time. As abdominal and back muscle RPE dynamics was found to behave differently, we have the opinion to share this finding with the community. We changed the manuscript, to put this issue more in the foreground.

- Abstract should end with a concise description about the possible usefulness of the proposed data.

Answer

The discussion section of the abstract has been modified accordingly.

- Please change the word "prognostic" with "predictive" throughout the manuscript. 

Answer

We have changed this accordingly

- The group names (good/bad /start/end) are generic, inadequate and need to be more clearly related to the scale they refer to. Please edit throughout the text.

Answer

This particular part has been rewritten completely.

-Line 44-45: do you mean in real time?

Answer

Yes, this has now been added.

- Line 55: only for endurance? not also for resistance exertion?

Answer

This has been added to the tesxt

- Line 73-74: Add at least two references.

Answer

We have added the respective references.

- Methods: for the exclusion criteria, wouldn't  be better to indicate any permanent or transitory condition that may affect the ability to carry out the proposed exercises?

Answer

We agree, that any condition – acute or permanent that prevents the subject to be investigated is by itself a precluding condition. On the other hand, we wanted to be as exact as possible, to mention also exclusion criteria due to simple technical reasons (device limits) and considerations according the also applied EMG investigation.

- Table 1: Each acronym used should have its extended form below the table

Answer

All acronyms are now mentioned below the table

- Investigation: please add a descriptive/representative image of the experimental set up and of the proposed exercises.

Answer

We have now added the new figure 1 showing the investigation situation.

- Figure 3 is unnecessary.

Answer

According to your suggestion we have now deletd this figure.

-Line 197-199: Why?

Answer

Its just as stated. We wanted to know, if reported RPE values at the beginning of a limited endurance test are of any predicitive value. As the results showed, that the development of RPE values differs between abdominal and back muscles we still feel this particular result worth to be communicated with the scientific community.

-Line 203-206: Please revise the statement in a less absolutist form.

Answer

Changed as recommended

-Line 211-212: Is this percentual universal? Please revise the statement.

Answer

The statement was revised as recommended.

- Line 217-218: It may not be hypothetical but, based on the proposed references (both including 39 subjects, same cohort?), it is not written in stone either! Please modify the sentence by specifying that the proposed concept is in line with previous studies. Also, please, if possible, it would be better to avoid too many self-citations.

Answer

First: the mentioned cohort is different from the actual one, but in several similar studies (not separately published) always these observations could be repeated, i.e. the non- linear amplitude to force relationship of abdominal vs. the almost ideal linear relationship for the back muscles. I agree, that self citations are not opportune, but to our knowledge, these results were only published by our group. So, simply there are no conflicting and also no confirming investigations available. The sentence was changed as recommended.

- Line 219-221: Please widely modify.

Answer

The respective section has been modified as recommended

- Line 222: delete "evidently".

Answer

Deleted as recommended.

- Line 230: Please add a reference to the text parts which report it.

Answer

The respective reference has now been added.

- Line 241: is this reported in the methods section?

Answer

We have now added the respective information in the methods.

- Weren't you aware of the issue? Why didn't you include a certain level of experience of the participants among the inclusion criteria to avoid this bias? This is a gross methodological error!

Answer

We made sure that the study participants in both sub-studies did not have extensive experience in recreational sports to exclude related influencing factors. This ensures that both subgroups are comparable and do not consist of participants with varying levels of athletic experience. We assessed the physical activity level before the investigation using a questionnaire. This information can be added if needed.-

Line 243-245: this is true for isometric, but not isotonic, task (according to the mentioned reference). Please reformulate.

Answer

We have changed the respective section as recommended

- Line 247-249: Is this data universally generalizable? Rephrase the sentence please.

Answer

The sentence was rephrased as recommended

-Line 250-252: Where is this mentioned in the methods section? Why of the opposite sex?

Answer

This particular information has now been added to the methods together with the respective justification

Reviewer 3 Report

Comments and Suggestions for Authors

Basic reporting

The authors present an original article in which they investigated the extent to which perceived exertion levels at the beginning, or the end of a time-limited endurance task of trunk muscles can be generalized, and whether values obtained at the beginning have prognostic value. Many aspects require clarification to enhance the manuscript's readability.

ABSTRACT

Please, include the main aim of the study.

Moreover, add the statistical methods used in your analysis and conclude the abstract with the main practical implications of your study.

Try to not repeat keywords which are include it in the title.

INTRODUCTION

The introduction is not well-structured and lacks essential information. It should be further developed to provide a clear context for your study.

L 66-74: This information should not be included here

MATERIAL AND METHODS

Participants

  • Please specify the training volume per week for the sample (in hours per week).
  • Improve the description of the sample selection process. How was the sample selected? Describe the method used to determine the sample size (such as G*Power) and whether the sample is representative. If it is, please introduce the sample size calculation and provide supporting references.

Investigation

·       I suggest including a figure of the test in order to help readers’ comprehension.

·       Please explain how and where the tests were conducted. Include details about the warm-up routines, who organized and administered the tests, and any other important information that helps the reader understand the testing process.

·       Were the tests randomized? If so, which procedure was used to randomize the order?

Statistical analysis

·       L 133-140: The sEMG information should been included in investigation section. Moreover  

RESULTS

·       General comment: There is a concern regarding the gender composition of the sample. Why are male and female not divided into separate groups? This can affect the results. Please clarify the reasoning behind this choice.

CONCLUSIONS

The conclusion section should be reorganized and further developed. Moreover, a practical implications section should be further developed. How can these findings help strength and conditioning trainers? Which practical implications can it have?

Author Response

The authors present an original article in which they investigated the extent to which perceived exertion levels at the beginning, or the end of a time-limited endurance task of trunk muscles can be generalized, and whether values obtained at the beginning have prognostic value. Many aspects require clarification to enhance the manuscript's readability.

ABSTRACT

Please, include the main aim of the study.

Answer

The study aim has been added now.

Moreover, add the statistical methods used in your analysis and conclude the abstract with the main practical implications of your study.

Answer

The respective information has been added as recommended

Try to not repeat keywords which are include it in the title.

Answer

The keywords have been modified as recommended

INTRODUCTION

The introduction is not well-structured and lacks essential information. It should be further developed to provide a clear context for your study.

Answer

We have largely revised the Introduction.

L 66-74: This information should not be included here

Answer

Typically, at the end of the introduction, a brief outlook on the study is provided, along with one or more formulated hypotheses or expectations regarding the results. Since the other reviewers did not criticize this section, we would like to keep it as it is.

MATERIAL AND METHODS

Participants

Please specify the training volume per week for the sample (in hours per week).

Answer

The respective information has been added as recommended.

Improve the description of the sample selection process. How was the sample selected? Describe the method used to determine the sample size (such as G*Power) and whether the sample is representative. If it is, please introduce the sample size calculation and provide supporting references.

Answer

The presented results are to be considered as partial results of two larger studies, which investigated reliability characteristics of SEMG measures during endurance tasks. As it turned out that the queried RPE values contain unexpected characteristics we decided to publish these particular results. Therefore, the sample size was calculated for repeated measures which require 34 subjects if two sided tests are applied (effect size 0.5, power 0.8). So, for the actual data no separate calculation was done. As with the application of the non-parametric U-test for independent samples plus the Bonferroni-Holm correction all obvious differences could be statistically proven the sample size can be considered sufficient. Additionally, we also calculated effect sizes, which are largely independent from sample size. In a kind of post-hoc proof their values justify the sample sizes of the subgroups. This information has been added to the respective part of the methods section.

A statement about the sample selection process has been added to the respective section.

Investigation

  • I suggest including a figure of the test in order to help readers’ comprehension.

Answer

The respective figure has been added

  • Please explain how and where the tests were conducted. Include details about the warm-up routines, who organized and administered the tests, and any other important information that helps the reader understand the testing process.

Answer

The respective information has now been added. The description of the investigation procedure has been modified.

  • Were the tests randomized? If so, which procedure was used to randomize the order?

Answer

The substudies were conducted at different times. Therefore no randomization was performed. A respective statement has been added.

Statistical analysis

  • L 133-140: The sEMG information should been included in investigation section. Moreover

Answer

The conducted SEMG measurement only serves as a supporting parameter and will be analyzed in detail elsewhere. Therefore we would like to leave them in place. Further, according to another reviewers suggestion we have restructured the methods section for clarity.

RESULTS

  • General comment: There is a concern regarding the gender composition of the sample. Why are male and female not divided into separate groups? This can affect the results. Please clarify the reasoning behind this choice.

Answer

The subgroup allocation was solely based on RPE levels at the respective time points. This is now clearly stated in the respective section of the methods.

CONCLUSIONS

The conclusion section should be reorganized and further developed. Moreover, a practical implications section should be further developed. How can these findings help strength and conditioning trainers? Which practical implications can it have?

Answer

We assume, that you mean the discussion. Incorporating your and also the comments of the other reviewers we have reorganized the discussion. Practical implications are now also mentioned explicitly.

Reviewer 4 Report

Comments and Suggestions for Authors

Thank you for submitting the paper for review.

1.      At the end of the introduction, please clearly include information about the study's objectives and hypotheses. To enhance the scientific quality, the authors may also add an anti-hypothesis.

2.      Please include sample size calculations - DOI: 10.1002/jgf2.600.

3.      In the context of statistics, please conduct a distribution analysis. Simply stating ‘Mann-Whitney U-test due to the small sample sizes of the subgroups’ is not sufficient.

4.      ‘Bonferroni-Holm correction to avoid alpha error accumulation’ – I am glad that the authors are using this correction, but please specify the exact p-value adopted by the authors in the text.

5.      ‘Additionally, we also calculated the corresponding effect sizes.’ – I am pleased that the authors are using effect sizes according to international recommendations. However, please include the ranges of these effects in the description. This will help readers better understand the study.

6.      Additionally, the authors studied sEMG. The description of sEMG is unacceptable and lacks many details. For example, how were the electrodes attached, how was the skin prepared, what electrodes were used, what was the maximum impedance, were the measurements repeated, how many times, was the ICC calculated, how was the signal cleaned? Please revise this thoroughly. What were the positions for MVC? The study must be replicable. Please refer to the reporting guidelines for sEMG studies: ‘The following supporting information can be downloaded at: https://www.mdpi.com/article/10.3390/jcm13051328/s1, Table S1. Recommendations for a list of elements that should be included in the description of the sEMG study.’ Please refer to the study: 10.3390/jcm13051328.

7.      The study limitations should be placed before the conclusions.

8.      Please prepare the bibliography according to the journal's guidelines – bold the year.

Author Response

  1. At the end of the introduction, please clearly include information about the study's objectives and hypotheses. To enhance the scientific quality, the authors may also add an anti-hypothesis.

Answer

The section was modified as recommended

  1. Please include sample size calculations - DOI: 10.1002/jgf2.600.

Answer

The presented results are to be considered as partial results of two larger studies, which investigated reliability characteristics of SEMG measures during endurance tasks. As it turned out that the queried RPE values contain unexpected characteristics we decided to publish these particular results. Therefore, the sample size was calculated for repeated measures which require 34 subjects if two sided tests are applied (effect size 0.5, power 0.8). So, for the actual data no separate calculation was done. As with the application of the non-parametric U-test for independent samples plus the Bonferroni-Holm correction all obvious differences could be statistically proven the sample size can be considered sufficient. Additionally, we also calculated effect sizes, which are largely independent from sample size. In a kind of post-hoc proof their values justify the sample sizes of the subgroups. This information has been added to the respective part of the methods section.

  1. In the context of statistics, please conduct a distribution analysis. Simply stating ‘Mann-Whitney U-test due to the small sample sizes of the subgroups’ is not sufficient.

Answer

We have now added a new table 2, containing all relevant data of the RPE- subgroups

  1. ‘Bonferroni-Holm correction to avoid alpha error accumulation’ – I am glad that the authors are using this correction, but please specify the exact p-value adopted by the authors in the text.

Answer

The global p- value was set p<0.05. This is now reported in the methods and also in the results section.

  1. ‘Additionally, we also calculated the corresponding effect sizes.’ – I am pleased that the authors are using effect sizes according to international recommendations. However, please include the ranges of these effects in the description. This will help readers better understand the study.

Answer

To our knowledge, there is no possibility to calculate confidence intervals for non-parametric effect sizes. We would happy to include them, if you provide the respective equation. If you mean the assessment of the fond ES values – they are already displayed below (new) Table 3.

  1. Additionally, the authors studied sEMG. The description of sEMG is unacceptable and lacks many details. For example, how were the electrodes attached, how was the skin prepared, what electrodes were used, what was the maximum impedance, were the measurements repeated, how many times, was the ICC calculated, how was the signal cleaned? Please revise this thoroughly. What were the positions for MVC? The study must be replicable. Please refer to the reporting guidelines for sEMG studies: ‘The following supporting information can be downloaded at: https://www.mdpi.com/article/10.3390/jcm13051328/s1, Table S1. Recommendations for a list of elements that should be included in the description of the sEMG study.’ Please refer to the study: 10.3390/jcm13051328.

The reporting of the sEMG data only serves as an additional supporting parameter to provide some information about how the muscles were stressed. As the article does not focus on sEMG, we therefore did not report the respective information completely. We have submitted two other manuscripts dealing with the sEMG issue: one about reliability characteristics (submitted to Journal of Electromyography and Kinesiology) and another one about fatigue- related issues (submitted to Journal of Sport and Health Science). We have now added some more global information about the sEMG measurements. If you feel still necessary to report the complete sEMG information we can provide the details.

  1. The study limitations should be placed before the conclusions.

Answer

The position was changed as recommended

  1. Please prepare the bibliography according to the journal's guidelines – bold the year.

Answer

In the submitted manuscript the year occurred in bold letters – this was lost by the production of the submission for review. At this point we cannot do anything.

Round 2

Reviewer 2 Report

Comments and Suggestions for Authors

I sincerely thank the authors for their efforts in editing the work. As a result, the work is clearer and of potential interest to the reader. 

However, I have to make some criticisms.

First of all, to facilitate the reviewer's work, it would be desirable to use the track changes function to clearly show the deleted parts while you have only highlighted the additions. Furthermore, if the corrected parts of the text no longer correspond to the number of lines in the original draft, it would be good practice to indicate the corresponding line number of the new version.

Here is the point by point section. All my responses are in bold:

Thank you for the opportunity to review this work. My main concern is the usefulness and potential application of these findings.  I don't see what the meaning of this study is, what is the novelty  that can drive the field vertically. This does not make me inclined to endorse the publication but I want to give the authors the opportunity to clarify this (fundamental) point. I choose the major revisions, for now, for the general clarity and correctness of the data analyses. However, the following points must be addressed.

Answer

We have completely revised the manuscript to improve language.

- The writing has improved, although not flawless.

What improvements do your results bring? How? The authors should propose practical applications of  the main result of the work (i.e. that it is possible to predict RPE values ​​for the abdominal muscles based on the initial RPE values). Please note that formally the work needs an extensive revision. First of all, an english native speaker revision in requested. Here some additional hints:

Answer

During submaximal endurance tasks premature failure is not due to physiological, but psychological, i.e. motivational reasons, i.e. the perceived exertion during such tasks. Therefore it makes sense to query such ratings during endurance tasks, to predict failure time. As abdominal and back muscle RPE dynamics was found to behave differently, we have the opinion to share this finding with the community. We changed the manuscript, to put this issue more in the foreground.

- I thank the authors for the answer but I find it unsatisfactory. It continues to be unclear to me how  this data can be useful, in a practical or research context. Please try to propose applications in these contexts.

- Abstract should end with a concise description about the possible usefulness of the proposed data.

Answer 

The discussion section of the abstract has been modified accordingly.

- Refer to line 31-35? This part does not address the question I raised as it offers an explanation of the results, but the usefulness or possible use is not highlighted.

- Please change the word "prognostic" with "predictive" throughout the manuscript. 

Answer

We have changed this accordingly

 - Ok

- The group names (good/bad /start/end) are generic, inadequate and need to be more clearly related to the scale they refer to. Please edit throughout the text.

Answer

This particular part has been rewritten completely.

- It seems to me that the definition is still present in the text. Please correct.

-Line 44-45: do you mean in real time?

Answer

Yes, this has now been added.

- Ok

- Line 55: only for endurance? not also for resistance exertion?

Answer

This has been added to the tesxt

- Ok

- Line 73-74: Add at least two references.

Answer

We have added the respective references.

- I see the addition of only one reference. By the way, the addition is not highlighted by the trackchanges function. If the changes are not clearly highlighted, the review process becomes very complicated.

- Methods: for the exclusion criteria, wouldn't  be better to indicate any permanent or transitory condition that may affect the ability to carry out the proposed exercises?

Answer

We agree, that any condition – acute or permanent that prevents the subject to be investigated is by itself a precluding condition. On the other hand, we wanted to be as exact as possible, to mention also exclusion criteria due to simple technical reasons (device limits) and considerations according the also applied EMG investigation.

- Ok

- Table 1: Each acronym used should have its extended form below the table

Answer

All acronyms are now mentioned below the table

- Good

- Investigation: please add a descriptive/representative image of the experimental set up and of the proposed exercises.

Answer

We have now added the new figure 1 showing the investigation situation.

- Good

- Figure 3 is unnecessary.

Answer

According to your suggestion we have now deletd this figure.

- Ok

 -Line 197-199: Why?

Answer

Its just as stated. We wanted to know, if reported RPE values at the beginning of a limited endurance test are of any predicitive value. As the results showed, that the development of RPE values differs between abdominal and back muscles we still feel this particular result worth to be communicated with the scientific community.

- Dear authors, my question is aimed at inducing you to highlight the usefulness of the work. It needs to emerge why you have studied this issue and how this results can bring improvements, whathever the field . Saying that you believe the results are worthy of sharing with the scientific community is not a satisfactory answer.

-Line 203-206: Please revise the statement in a less absolutist form.

Answer

Changed as recommended

- Ok

-Line 211-212: Is this percentual universal? Please revise the statement.

Answer

The statement was revised as recommended.

- Good

- Line 217-218: It may not be hypothetical but, based on the proposed references (both including 39 subjects, same cohort?), it is not written in stone either! Please modify the sentence by specifying that the proposed concept is in line with previous studies. Also, please, if possible, it would be better to avoid too many self-citations.

Answer

First: the mentioned cohort is different from the actual one, but in several similar studies (not separately published) always these observations could be repeated, i.e. the non- linear amplitude to force relationship of abdominal vs. the almost ideal linear relationship for the back muscles. I agree, that self citations are not opportune, but to our knowledge, these results were only published by our group. So, simply there are no conflicting and also no confirming investigations available. The sentence was changed as recommended.

- I find the response on the articles cited convincing but not the change made. I would change the sentence "This is not only hypothetical but is confirmed in existing studies" with the sentence "This seems to be in line with some existing studies" and I would delete the sentence you added on line 248-249.

- Line 219-221: Please widely modify.

Answer

The respective section has been modified as recommended

- This part should be rewritten in a less absolutist key.

- Line 222: delete "evidently".

Answer

Deleted as recommended.

- Ok

- Line 230: Please add a reference to the text parts which report it.

Answer

The respective reference has now been added.

 - Please add a cross reference to the part of the results that supports this sentence:  "Both linearly interpolated RPE curves showed similar inclinations, so that regardless of  the reference time point, the gap between the two RPE levels remained almost constant over the 10-minute test period." line 260-262.

- Line 241: is this reported in the methods section?

Answer

We have now added the respective information in the methods.

 - Ok

- Weren't you aware of the issue? Why didn't you include a certain level of experience of the participants among the inclusion criteria to avoid this bias? This is a gross methodological error!

Answer

We made sure that the study participants in both sub-studies did not have extensive experience in recreational sports to exclude related influencing factors. This ensures that both subgroups are comparable and do not consist of participants with varying levels of athletic experience. We assessed the physical activity level before the investigation using a questionnaire. This information can be added if needed.-

 I really appreciate the rational but if, as you wrote, inexperienced participants tend to underestimate the level of demand at the beginning, why did you make sure that the study participants in both sub-studies did not have  experience in recreational sports? Isn't this like intrinsically introducing a bias in the results? Please argue this.

Line 243-245: this is true for isometric, but not isotonic, task (according to the mentioned reference). Please reformulate.

Answer

We have changed the respective section as recommended

- Ok

 - Line 247-249: Is this data universally generalizable? Rephrase the sentence please.

Answer

The sentence was rephrased as recommended

- Please change "proven" with "reported". Once again, the change to the text is not highlighted.

-Line 250-252: Where is this mentioned in the methods section? Why of the opposite sex?

Answer

This particular information has now been added to the methods together with the respective justification

- Line 144: The methodology should be standard. "Verbal encouragement was also provided, if considered necessary" is not a good phrase. So not all subjects were supported and encouraged? On what basis was verbal encouragement deemed necessary for some? When was it not considered as such? Furthermore, is there any data to support motivational bias due to gender?

Overall, I am not very satisfied with the responses received. I invite the authors to respond formally and substantially more adequately next time.

Comments on the Quality of English Language

Many errors are still present.

Reviewer 3 Report

Comments and Suggestions for Authors

nc

Author Response

We tnak the reviwewer for the positive evaluation of the revision

Round 3

Reviewer 2 Report

Comments and Suggestions for Authors

The authors responded satisfactorily to my requests.

Author Response

We thank the revewer for the positive evaluation of the revision.